# Reassessment of Surgical Procedures for Complex Obstructive Genital Malformations: A Case Series on Different Surgical Approaches

**DOI:** 10.3390/jcm11175026

**Published:** 2022-08-26

**Authors:** Alice Hoeller, Sahra Steinmacher, Katharina Schlammerl, Markus Hoopmann, Christl Reisenauer, Valerie Hattermann, Sara Y. Brucker, Katharina Rall

**Affiliations:** 1Department for Womens’ Health, University of Tuebingen, 72074 Tübingen, Germany; 2Department for Radiology, University of Tuebingen, 72074 Tübingen, Germany

**Keywords:** obstructive Müllerian duct malformation, cervical aplasia, cervical atresia, hematometra, uterovaginal anastomosis

## Abstract

The objective of this case series was to describe different uterus-preserving surgical approaches and outcomes in patients with complex obstructive Müllerian duct malformation caused by cervical and/or vaginal anomalies. A retrospective analysis was performed including patients undergoing uterovaginal anastomosis (n = 6) or presenting for follow-up (n = 2) at the Department for Gynecology at the University of Tuebingen between 2017 and 2022. Uterovaginal anastomosis was performed with a one-step combined vaginal and laparoscopic approach (method A), a two-step/primary open abdominal approach with primary vaginal reconstruction followed by abdominal uterovaginal anastomosis after vaginal epithelization (method B) or an attempted one-step approach followed by secondary open abdominal uterovaginal anastomosis due to reobstruction (method A/B). Patients presented at a mean age of 15 years. Two patients were treated by method A, four by method B and two by method A/B. Functional anastomosis was established in seven of eight patients, with normal vaginal length in all patients. Concerning uterovaginal anastomosis, the primary open abdominal approach with or without previous vaginal reconstruction seems to have a higher success rate with fewer procedures and should be implemented as standard surgical therapy for complex obstructive genital malformations including the cervix.

## 1. Introduction

Obstructive genital malformations including cervical agenesis/aplasia with or without transverse vaginal septum or (partial) vaginal aplasia belong to the group of complex Müllerian duct developmental disorders. In order to understand genital development disorders, it is essential to gain insight into the fundamentals of female embryological development. The genetic sex of each individual is determined by fertilization. In early embryological development (week 5–8), the mesonephric (Wolffian) and paramesonephric (Müllerian) ducts develop simultaneously. However, the presence of anti-Müllerian hormone (AMH), which is produced in the Sertoli Cells of the developing testis, disrupts the further development of the Müllerian duct in males. In absence of AMH, the Müllerian duct develops into the female phenotype, forming the uterus, cervix, fallopian tubes and the proximal part of the vagina. This process includes the elongation, fusion, canalization and partial resorption of the Müllerian duct. At approximately week 12 of gestation, the distal Müllerian duct fuses with the urogenital sinus, forming the vagina. Each of these steps is crucial for the unremarkable development of the female internal genitalia, and disruption at any point can cause different types of genital developmental disorders, including cervical agenesis or aplasia [1,2,3]. In general, Müllerian duct anomalies are described in 4–7% of the female population, although cases of cervical agenesis or aplasia (with or without a vagina) but with a functional uterus are rare [4,5,6,7]. 

Young women or girls often initially present with abdominal pain, primary amenorrhea and/or incapacity for sexual intercourse in cases of associated vaginal aplasia or thick transverse septum. In the long term, the incapacity for orthograde menstruation can result in hematometra/-kolpos with medical complications such as retrograde menstruation leading to endometriosis, organ adhesions, severe abdominal pain, infertility and infection. As a temporary solution, hormonal suppression of menstruation might be induced to prevent the reoccurrence of hematometra until a surgical strategy has been decided on or a functional uterovaginal anastomosis has been established. Such an approach can be reasonable for very young girls, as compliance with and understanding of therapy is essential. The limited surgical expertise and a lack of standardization with respect to the formation of a functional uterovaginal anastomosis or reformation of a functional cervix have historically led to hysterectomy as a radical surgical approach in young women or girls [8,9,10]. Nowadays, gynecological specialists seek not only to reduce morbidity but also to conserve the possibility of reproductive capacity. Therefore, there have been some reports describing surgical management [6,10,11]. In this article, we present different surgical approaches to uterus-conserving surgical therapy for different Mullerian anomalies performed at the Department for Women’s Health of the University of Tuebingen, Germany, between 2017 and 2022.

## 2. Materials and Methods

This retrospective analysis included a total of 8 patients; 6 patients underwent surgery for uterovaginal anastomosis from 2017 to February 2022 at the Department for Women’s Health of the University of Tuebingen, Germany. Two additional patients who presented at our outpatient clinic for genital malformations for follow-up examination within this period of time but had surgery performed in the years 2012 and 2015, respectively, were also included. 

Mullerian duct anomalies were classified using the ESHRE/ESGE consensus classification system from 2013 (European Society of Human Reproduction (ESHRE) and Embryology-European Society for Gynaecological Endoscopy (ESGE) [5,12]. Patients included in this retrospective study presented with obstructive genital malformation due to cervical aplasia or atresia (C4), unilateral cervical aplasia (C3) and/or vaginal aplasia (V4) or transverse vaginal septum (V3).

Patient information concerning medical history, diagnosis, symptoms and treatment were obtained from medical records within the patient data system SAP GUI 7.70_DE as of January 2021. To depict the heterogenicity of the analyzed cases, a brief case report was summarized for each patient. Charts and tables were generated using Microsoft^®^ Excel^®^ 2019 and Microsoft^®^ Word^®^ 2019 (Redmond, WA, USA). Due to the descriptive character of the present study, no statistical tests were performed.

Diagnostic and surgical strategy:

The diagnostic algorithm consisted of a thorough gynecologic examination, including the inspection of the vulva and, if possible, the vagina, vaginal and abdominal palpation; a vaginal, transabdominal or perineal 2D and 3D ultrasound; an MRI scan of the pelvis; and, when needed, laparoscopic validation of genital tract anomalies. 

Patients with vaginal aplasia or thick obstructing proximal vaginal septum underwent treatment for neovagina/vaginal reconstruction following the modified McIndoe technique without a mesh graft, creating a vaginal canal by sharp and blunt dissection between the bladder and rectum [13,14,15]. Uterovaginal anastomosis was either laparoscopically established in a one-step procedure similar to the pull-through technique described elsewhere [16] (method A) or in a two-step/primary open abdominal procedure. In the latter cases, if necessary, a neovagina was first created as previously described or vaginal reconstruction/resection of vaginal septum was performed. After complete epithelialization of the neovagina, as a second step, a uterovaginal anastomosis was created using a combined vaginal and open abdominal approach similar to the push-trough method described elsewhere [16] (method B). Patients presenting with a normal vagina in width and length [17] were assigned to the two-step/open abdominal technique (method B) (Figure 1). 

In the remaining cases, we initially attempted a single-step uterovaginal anastomosis, but after recurring stenosis/obstruction, uterovaginal anastomosis was established in an open abdominal approach (method A/B) similar to that adopted in method B. Patients presenting before 2017 were initially treated by method A, with the approach shifting toward method B after September 2019 as a result of surgical experience. The surgical treatment algorithm is shown in Figure 1. Ureters were preoperatively splinted prophylactically. Surgery was performed under a maximum of safety precautions, including vaginal, rectal and abdominal palpation and cystoscopy and intraoperative intravenous antibiotic application. Short-term outcomes were mostly evaluated in a secondary diagnostic hysteroscopy and/or gynecologic examination 4–6 weeks after surgery for uterovaginal anastomosis.

Definition of outcome parameters:

Outcome measures were defined as follows:

1.Surgical outcome parameters:
1.1Primary surgical strategy1.2Number of surgeries in total

Diagnostic or, if required, unplanned surgery for the creation of a neovagina/vaginal reconstruction, uterovaginal anastomosis, planned follow-up examination with cervical dilation and hysteroscopy, recurrent cervical dilation, secondary uterovaginal anastomosis and other kinds of secondary surgery were performed.

2.Clinical outcome parameters concerning vagina and anastomosis
2.1Vaginal length and sexual intercourse;2.2Orthograde menstruation;2.3Sonographic exclusion of hematometra/-kolpos;2.4Restenosis or reoccurring hematometra; and2.5Complications.

## 3. Results and Presentation of Cases

### 3.1. Patient Characteristics

Patient characteristics are summarized in Table 1. Patients presented at a mean age of 15 years (range 11–20 years). Indications for clinical presentations at our department were either recurrent pelvic or abdominal pain due to hematometra or, in one case, pyometra, diagnostics for primary amenorrhea or the request for further treatment after diagnosis and/or treatment at another hospital. 

There was one case of suspected OHVIRA (Obstructed hemivagina with ipsilateral renal anomaly) with bicorporal uterus (U3b) with unilateral cervical atresia (C3) and thick proximal transverse vaginal septum (case 1); three cases of hemiuterus (U4), either with (U4a; n = 2) or without (U4b; n = 1) rudimentary cavity; three cases of normal uterus (U0); and one case of dysmorphic T-shaped uterus (U1a). Complete cervical atresia (C4) was present in five cases, with cervix duplex (C3) in one case and two cases of patients with a normally configured cervix (C0) but vaginal aplasia (V4) or thick proximal transverse vaginal septum (V3). Vaginal aplasia (V4) was reported in two cases, with a transverse vaginal septum (V3) in two cases. In two cases, differentiation between thick vaginal septum and partial vaginal aplasia was not possible (V3/V4), and a normal vagina (V0) was observed in two cases (Table 1). Four of the eight patients presented with previously diagnosed associated malformations ranging from renal, skeletal, complex anogenital to cardiac defects. Five patients had been diagnosed with endometriosis genitalis externa.

### 3.2. Surgical Outcome Parameters

Surgery was performed at the Department for Women’s Health of the University of Tuebingen, Germany. Each Surgery for uterovaginal anastomosis was performed by a team of at least two gynecologists with expertise in vaginal, laparoscopic and open abdominal surgery specialized in female genital malformations; one of the gynecologists was present at every operation. The continuity of the surgical team ensured that the outcome was not altered by expertise or surgical abilities. 

The surgical strategy is visualized in Figure 1, and surgical outcome parameters are summarized in Table 2. In six of eight patients, surgical treatment for neovagina was achieved using the modified McIndoe technique, or vaginal reconstruction/resection of vaginal septum was performed. In four cases, a neovagina was created in a single-step technique in combination with uterovaginal anastomosis (Cases 1–4). Two patients were treated by method A, four by method B and two by method A/B (Figure 1). The total number of surgeries ranged from 3 to 11. Planned follow-up surgery by means of hysteroscopy and cervical dilation examination was performed in six of the eight cases. Secondary surgery was required in cases 1–4. Patients treated with method A had a mean number of 6.5 surgeries, patients treated by method B had a mean number of 3.75 surgeries and patients treated with method A/B had an average of 9 surgeries (Table 2).

### 3.3. Clinical Outcome Parameters

Details are summarized in Table 3. All patients had a normal vaginal length > 6 cm [17], and three patients reported having had satisfying sexual intercourse.

The absence of hematometra as an outcome parameter for functionality of uterovaginal anastomosis was achieved in all patients, although there was one case of recurrent pyometra, leading to hemi-hysterectomy and preserving the functional hemiuterus (case 1). Both patients treated by method A reported regular menstruation, with one patient menstruating out of the normally functioning hemiuterus (Case 1). One patient treated by method A/B reported regular menstruation, whereas the other patient did not menstruate under progesterone-only pill for contraception. The menstrual history of the four patients treated with method B is unknown/pending; one patient was lost to follow-up, and three patients had only received surgical treatment within the last three months of writing this article. Reasons for secondary surgery were reoccurrence of cervical or vaginal stenosis and/or reobstruction. Among all patients, there was the one case of hemi-hysterectomy, as mentioned above (case 1). No other complications occurred apart from the need for re-surgery due to stenosis and/or obstruction. To date, there has been no reported case of pregnancy.

### 3.4. Presentation of Cases 

#### 3.4.1. One-Step Approach, Method A

Case 1

A 17-year-old girl presented in April 2015 with symptomatic advanced hematometra and partial hematokolpos. Clinical examination and ultrasound showed a bicorporal uterus with hematometra of the right side with suspected cervical hypoplasia and thick proximal vaginal septum with proximal hematokolpos on the right side. The left cervix and uterus seemed to be unaffected (ESHRE/ESGE classification: U3b C3 V3). Furthermore, the patient showed an aplasia of the right kidney; therefore, OHVIRA (obstructed hemivagina with ipsilateral renal anomaly) syndrome was suspected. Old blood was drained after dissection of the thick horizontal vaginal septum and reconstruction of the vagina with a method similar to the pull-through technique described by Bijsveldt et al., [16] thereby creating a uterovaginal anastomosis. Hysteroscopy of both hemiuteri was unsuspicious, although the right cervix could not be clearly detected. To prevent adhesions and closure, a Foley catheter was placed within the anastomosis. On the 8th day after surgery, the Foley catheter was replaced by a Fehling tube. One month later, the Fehling tube was removed, and cervical dilation was conducted. In August 2015, approximately 3 months after the primary surgery, the patient presented with recurrent hematometra on the right. Cervical dilation, hysteroscopy and diagnostic laparoscopy were performed, and again, a Foley catheter was placed into the cervix. The catheter was removed 5 days later, and the cervix was fixated by sutures. Due to recurrent hematometra on the right and physiological function of the left uterus, in October 2015, a right-sided supracervical hemihysterectomy was performed laparoscopically. With vaginal examination under general anesthesia prior to hysterectomy, we were not able to detect the previously reconstructed right cervix. The patient presented for follow up 1 month and 3 months after hysterectomy. She was asymptomatic and menstruated regularly, and the vaginal length was described as unsuspicious. Sonography showed minimal retention of fluid in the region of the previously formed right cervix. In July 2016, the patient was admitted due to abdominal pain and sonographically suspected with a hematocervix of 52 × 29 × 32 mm. Cervical mucus relief was obtained by laparoscopic incision. Again, the right-sided cervicovaginal anastomosis was not patent, cervical dilation was performed and a Fehling tube was inserted. Due to recurrent mucocervix, the cervix was surgically removed in December 2018. No further consultations at our department have taken place since then.

Case 2

A 13-year-old girl presented with acute abdominal pain; therefore, emergency laparoscopy was performed. In addition to the medical history of multiple malformations such as bilateral double kidneys, skeletal malformations, anal atresia and tetralogy of fallot—for which surgery had been performed previously—laparoscopy showed vaginal and cervical atresia with a considerably enlarged right-sided hemiuterus and a rudimentary left uterine horn (ESHRE/ESGE classification: U4a/C4/V4), as depicted in Figure 2. After transfundal incision, pus was discharged. Then, a neovagina was created using the modified McIndoe procedure, and within the same surgery, a vaginally assisted laparoscopic uterovaginal anastomosis was established and a catheter was placed to contain the anastomosis. Five days later, the catheter was removed, and hysteroscopy and laparoscopy and reconstruction of the cervix were performed. A Fehling tube was inserted into the newly formed cervix, and the patient was instructed to use a vaginal stent at night. One month later, in March 2012, although the Fehling tube was still in the correct position within the uterovaginal anastomosis with no sign of hematometra, the vagina had obliterated completely and had to be reopened. Although a cervical formation had been performed in earlier surgeries, a cervix could not be identified. The Fehling tube was left within the anastomosis, and a vaginal device was placed into the vagina. Two months later, the patient presented for the removal of the vaginal device and Fehling tube. The neovagina was completely epithelialized and showed no sign of adhesions. However, the previously created cervix could now be identified as such. Aside from a small cervical polyp, hysteroscopy showed a normal cervical canal and hemiuterus. Follow-up examinations 1 and 3 months later showed no sign of hematometra, and menstruation occurred regularly. In November 2012, the patient was admitted due to pyometra for antibiotic and surgical treatment. The vagina had obliterated to a length of 2 cm, and the cervix was constricted. After dilation, pus discharged from the uterus. Vaginal reconstruction was performed, and a vaginal phantom was inserted. Follow-up examinations showed no sign of hematometra, and regular menstruation was achieved under oral hormonal contraceptive therapy. In September 2013, the patient was admitted again for therapy of pyometra. Laparoscopy and hysteroscopy showed advanced adhesions, which were cleaved. Follow-up 3 months later showed no sign of hemato- or pyometra, although menstruation was temporarily suppressed by hormonal therapy. The last presentation at our clinic was in 2018; menstruation occurred regularly, sexual intercourse was satisfying with a vaginal length of 7 cm and there was no sign of hemato- or pyometra.

#### 3.4.2. One-Step Approach with Secondary Abdominal Uterovaginal Anastomosis Due to Reobstruction, Method A/B

Case 3

A 12-year-old girl presented for further therapy after surgery was performed in another hospital due to severe abdominal pain with intraoperative diagnosis of partial vaginal aplasia or thick horizontal vaginal septum and hematometra with a left hemiuterus (ESHRE/ESGE classification: U4b C0 V3/4). Associated malformation: agenesis of right kidney. In January 2015, we performed surgery to create a neovagina using the modified McIndoe technique. The assumed cervix seemed to be dilated due to hematometra; further dilation and hysteroscopy were performed under laparoscopic view. To maintain uterovaginal anastomosis, a catheter was placed into the cervix. Five days later, the intrauterine catheter was removed, and the patient was instructed how to use a vaginal phantom. Six months later, vaginal shorting and cervical stenosis with recurrent hematometra was diagnosed, dilation was performed, and a catheter was inserted into the cervix again. After removal of the catheter hematometra reoccurred; therefore, hormonal therapy was started to suppress menstruation. In spite of hormonal suppression, in December 2015, hematometra had to be drained as a result of recurrent vaginal shortening and stenosis, and a stent was placed into the vagina. Hormonal suppression of menstruation was continued, and although vaginal shortening and stenosis reoccurred, the patient remained asymptomatic. In March 2020, the patient asked for vaginal reconstruction. Due to vaginal constrictions and adhesions with only mild hematometra and the absence of sexual activity, vaginal reconstruction was abandoned at that time. Hormonal therapy was stopped in order to develop hematometra, and surgery was scheduled in September 2020. Surgery was performed by a combined vaginal and open abdominal approach. The vagina was opened from the abdomen, and the recurrent vaginal stenosis was solved by a vaginal push-through anastomosis. Again, the patient was instructed how to use a vaginal phantom. Follow-up visits took place regularly; no further vaginal stenosis occurred, and cohabitation was possible without dyspareunia, with a vaginal length > 12 cm. No hematometra was documented, although the patient was under hormonal therapy; therefore, menstruation was suppressed temporarily. The latest visit at out department was in November 2021 due to excessive vaginal discharge probably caused by the long-lasting sutures, which was treated locally; otherwise, the patient did not have any complaints. Due to a progesterone-only pill for contraception, no menstruation was reported.

Case 4

A 14-year-old girl presented with recurrent hematometra. The patient had been diagnosed with a bicorporal uterus with one cervix and a vaginal septum at another department of gynecology (ESHRE/ESGE classification: U4a C4 V3). Previous surgery at a different hospital: vaginal or hymenal incision and cervical dilation with discharge of hematometra and insertion of a catheter in the cervix. In a second-step vaginal reconstruction, laparoscopic removal of the left uterine horn had been performed, and a vaginal device and a Fehling tube were placed.

We initially performed surgery on the patient in March 2018. The vagina was nearly completely constricted; therefore, laparoscopic transfundal drainage of the hematometra was conducted. Then, a neovagina was reconstructed, and a vaginal device was put in place. Surgery for uterovaginal anastomosis was delayed until the vagina had healed completely; therefore, suppression of menstruation was initiated by hormonal therapy. With recurrent hematometra, in July 2019, vaginally assisted abdominal uterovaginal anastomosis was performed, and a catheter was placed into the anastomosis. One month later, vaginal dilation and dilation of the uterovaginal anastomosis was performed due to stenosis, and the patient was instructed to use a vaginal device again. In the following months, the patient presented regularly for follow-up, the vagina showed a circular stenosis with a length of 3–4 cm and menstruation was normal without signs of hematometra. In July 2021, 2 years after abdominal uterovaginal anastomosis, dilation of the uterovaginal anastomosis was necessary. The latest follow up was in March 2022; the patient showed no signs of hematometra, with a normal vaginal length and no signs of vaginal or uterovaginal stenosis.

#### 3.4.3. Two-Step/Open Abdominal Approach, Method B

Case 5

A 16-year-old asymptomatic girl presented with primary amenorrhea and suspected transverse vaginal septum or partial vaginal aplasia in October 2019. Physical examination showed an estimated vaginal length of 3 cm and no evidence of hematometra or -kolpos. Laparoscopy showed a normally configured uterus, intrabdominal adhesions and endometriosis genitalis externa (ESHRE/ESGE classification: U0 C0 V4). A neovagina was created using the modified McIndoe technique. Seventeen months after the successful creation of a neovagina, a cervicovaginal anastomosis was formed by vaginally assisted Pfannenstiel laparotomy similar to the push-through technique. A catheter was inserted into the newly created anastomosis. On the 6th postoperative day, the catheter was removed. The patient was instructed to insert a vaginal device. Within the next 8 months, five minor surgical interventions were performed for resolution of cervical stenosis. The last follow-up examination in January 2022—10 months after cervicovaginal anastomosis—showed a normal vaginal length and no signs of hematometra; however, menstruation was not reported since December 2021. Hormone analysis was unremarkable, and the endometrial thickness was 13 mm. At that time, surgical intervention was not indicated, and follow-up was recommended after 3–6 months or in case of any problems.

Case 6

A 20-year-old woman presented with a surgically assured diagnosis of cervical aplasia/atresia and endometriosis (ESHRE/ESGE classification: U0 C4 V0). At presentation, the patient was under medication with a progesterone-only pill. The patient was asymptomatic, and physical examination showed no sign of hematometra, with a normal vaginal length (9–11 cm). In preparation for surgery, the patient was weaned from hormonal medication and subsequently developed abdominal pain and hematometra (Figure 3). Surgery for uterovaginal anastomosis was performed by Pfannenstiel laparotomy using the push-through technique and resection of the atretic cervical tissue in February 2022 (Figure 3). The previously described endometrioses was no longer detected. Follow-up surgery was scheduled after 6 weeks, comprising a hysteroscopy without any sign of cervical stenosis (Figure 3). Further follow-up is anticipated in the future.

Case 7

A 17-year-old patient presented with presumed genital malformation. Previously, the patient had undergone surgery for presumed vaginal septum, and a cervix could not be found. The patient had initially presented to her gynecologist at the age of 15 year with primary amenorrhea and moderate cyclic pain. The clinical examination depicted a vaginal length of 5–7 cm with a blind ending. An MRI scan suggested a T-formed uterus, cervical atresia, vaginal septum and enlarged ovaries with cysts on both sides (Figure 4). Diagnostic surgery was performed to identify the exact extent of genital malformation. Laparoscopically, endometriosis could be assured, the uterus presented with a T shape, the cervix uteri seemed to be absent and transverse vaginal septum or partial proximal vaginal aplasia was suspected (ESHRE/ESGE classification: U1a C4 V3/4). In February 2022, uterovaginal anastomosis via Pfannenstiel laparotomy using the push-through technique and resection of the atretic cervical tissue were performed, and a 14 Ch-silicon catheter was inserted into the cervix (Figure 4). Follow-up hysteroscopy and cervical dilation after 5 weeks showed a normal vaginal length with sufficient anastomosis and confirmed the presumed T-shaped uterus with no sign of infection. The next examination for follow-up is scheduled in July 2022.

Case 8

An 11-year-old girl presented with severe abdominal pain and primary amenorrhea. Gynecological examination was performed under general anesthesia combined with diagnostic laparoscopy. The patient was found to have a vaginal length of 10 cm, but the cervix uteri was missing (ESHRE/ESGE classification: U0 C4 V0). Whereas the uterus appeared without hematometra or other abnormalities, a significant amount of blood was located in the abdomen, suggesting retrograde menstruation. Low-stage endometriosis was treated by coagulation. Due to the young age of the patient, hormonal therapy for suppression of menstruation was agreed upon. Regular follow-up was carried out in our outpatient clinic. At 14 years of age, the girl was admitted to hospital, and laparoscopy was performed because of severe abdominal pain and hematometra following reduced adherence to hormonal therapy. At 15 years of age, in September 2017, uterovaginal anastomosis was performed via Pfannenstiel laparotomy using the push-through technique, along with resection of the atretic cervical tissue, and a 14 Ch-silicon catheter was inserted into the newly formed anastomosis. The follow-up 4 weeks after surgery demonstrated an open cervix, and menstruation had not occurred, with no signs of hematometra or infection. The patient was then lost to follow-up.

## 4. Discussion

Due to the complex regulatory steps during Mullerian duct development, which include midline fusion, partial resorption and fusion with the urogenital sinus, development disorders can present in many different phenotypes. As identification of various types of obstructive Mullerian duct development is crucial for the success of surgical therapy, the combination of clinical examination, 2D and 3D ultrasound, MRI scanning of the pelvis and, in a second step, combined hysteroscopy and laparoscopy is the current diagnostic standard and should be performed before scheduling surgery for uterovaginal anastomosis [5,9].

In this case series, we describe eight cases of patients presenting with complex obstructive malformation including cervical aplasia or atresia with or without corresponding vaginal aplasia or thick proximal transverse vaginal septum. Unique medical histories with a diversity of symptoms or previously performed surgery make comparison of the cases difficult. Although discrimination between a thick proximal transverse vaginal septum and partial vaginal aplasia was not always possible [16], vaginal reconstruction and/or creation of a neovagina was necessary in both cases. 

In four patients, a single-step vaginal reconstruction with simultaneous primary laparoscopy similar to the pull-through method for vaginal septum resection, if necessary, combined with cervical canalization or drilling, was performed to create a uterovaginal anastomosis [16,18]. Surgeries were performed between 2012 and 2017. In cases 3 and 4, recurrence of uterovaginal or vaginal stenosis led to secondary surgery similar to the abdominal push-trough method [16] for uterovaginal anastomosis in 2019 and 2020, respectively.

Cases 5–8 underwent primary surgery between 2017 and 2022. If creation of a neovagina, resection of vaginal septum or vaginal reconstruction was necessary, a two-step procedure was conducted. First, a neovagina was created, and only after completed epithelization of the vagina, uterovaginal anastomosis was constructed similar to the abdominal push-through technique. Formation of a neovagina or vaginal reconstruction was necessary in two cases; the other two cases had no pathology of the vagina [19,20] and an open abdominal uterovaginal anastomosis was created (step two of the two-step procedure) (Figure 1).

With increasing surgical experience and insufficient patency of uterovaginal anastomosis in a single-step laparoscopic approach (Cases 1, 3, 4), a drift to a two-step and open abdominal approach with assumed advantages might be proposed.

A long-term analysis by Deffarges et al. (2001) depicts similar results [21]. In our opinion, two main factors might play a role in the outcome.

First, patients with vaginal aplasia or thick transverse proximal vaginal septum and cervical aplasia require two surgical procedures consisting of the creation of a neovagina/vaginal reconstruction and the formation of a uterovaginal anastomosis. Both surgeries are complex procedures. The neovagina or reconstructed vagina has to heal, and occlusion is routinely prevented by the use of a vaginal and or cervical insert without a mesh graft [13,22,23]. When both steps of surgery are performed simultaneously, risk of occlusion or stenosis of the neovagina and uterovaginal anastomosis might be aggravated [9]. This might have accounted for cases 1, 3 and 4, in which a functional anastomosis failed to be established in a simultaneously vaginal reconstruction/creation of neovagina and laparoscopic uterovaginal anastomosis. Risk of vaginal reobstruction might be increased, as patients with obstructive genital malformations present at a young age, and compliance with the use of a vaginal phantoms or stents might not be optimal in comparison to older patients presenting for planned surgery for a neovagina, for example, in cases of MRKH [19,20,22].

Secondly, the main advantage of open abdominal uterovaginal anastomosis similar to the push-through technique, which was performed in the suggested two-step/open abdominal approach of cases 4–8, is the possibility of adequate creation and mobilization of the atretic cervix or uterus and its suture to the proximal vagina in a circular manner, forming a steady anastomosis using non-resorbable sutures. A similar approach was described by Grimbizis et al. (2004), whereas Kraim et al. (2020) used absorbable sutures, although the authors did not comment on the exact implementation of anastomosis [18,24]. Furthermore, if necessary, a cone can be resected if a fibrous cord or atretic cervix is present, as previously described [25,26]. This extent of uterine/cervical mobilization and visualization does not seem to be laparoscopically possible. Thus, in laparoscopic approaches a (re)canalization of an atretic cervix or fibrous cord was aspired, and attempts to avoid restenosis were made by using Foley catheters or Fehling tubes.

Regarding the year of primary surgery, within the analyzed collective at the Department for Women’s Health of the University of Tuebingen, Germany, the primary surgical approach has changed from a single-step laparoscopic, to a one- or two-step open abdominal technique for uterovaginal anastomosis. This is in agreement with the current international recommendations [2,9,26].

Comparison of surgical strategies and the number of required surgeries is difficult, as literature is scarce and sole case reports are often presented [7,10,11,24,25,27,28]. Minami et al. (2019) described two cases of vaginal and cervical aplasia, which were successfully treated with the simultaneous creation of a neovagina using the modified McIndoe’s technique (using an artificial skin graft) and either abdominal or laparoscopic creation of a uterovaginal anastomosis [11].

In patients with OHVIRA syndrome, a vaginal septum dissection is often sufficient, although hemi-hysterectomy has also been reported as a surgical solution, especially in cases of thick proximal vaginal septum, as in the case of our patient (case 1) [29,30].

Limitations of this analysis include the small sample size and the fact that the there had been no long-term follow-up for cases 6–8 to date due to the recent surgical therapy. Because many patients presented at young age, evaluation of sexual intercourse or pregnancies were inconclusive, and further follow-up is needed.

## 5. Conclusions

In conclusion, cases of obstructive genital malformations can be complex and heterogenous. Thus, individual approaches are necessary. The surgical two-step technique involving the creation of a neovagina or vaginal reconstruction in cases of thick transverse vaginal septum and the interval abdominally modified push-trough surgery for uterovaginal anastomosis seems to have advantages in terms of the overall outcome and should therefore be preferred. Adequate therapy has to be realized in specialized centers engaging multiprofessional teams.

## Figures and Tables

**Figure 1 jcm-11-05026-f001:**
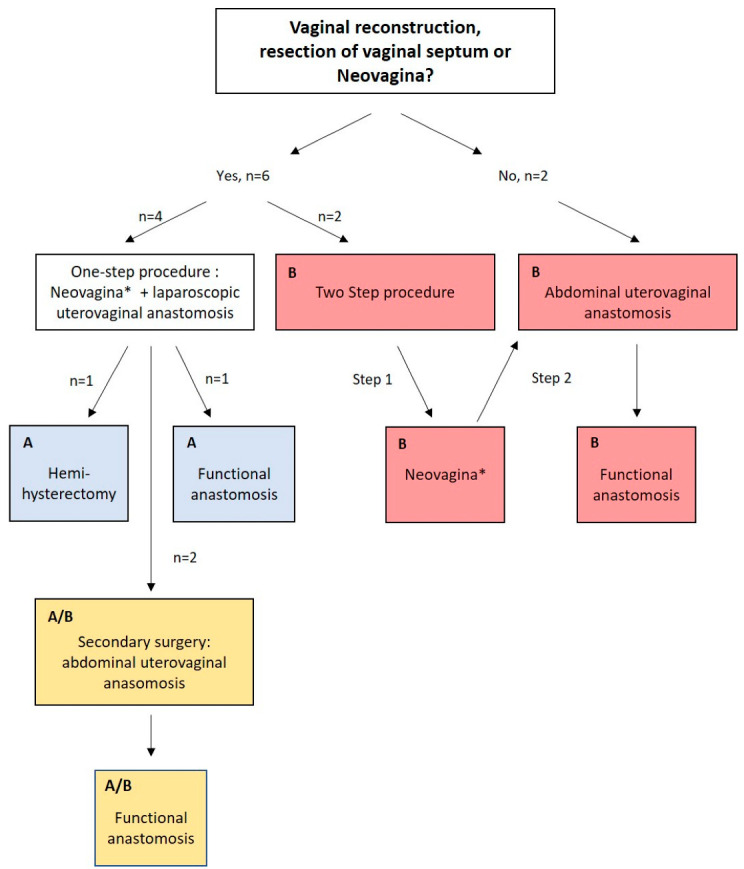
Surgical treatment algorithm. * Surgery for neovagina by modified McIndoe technique, vaginal reconstruction or resection of vaginal septum. (A) Method A: one-step creation of a neovagina* and simultaneous laparoscopic uterovaginal anastomosis procedure similar to the pull-through technique. (B) Method B: two-step/primary open abdominal technique: if necessary, creation of a neovagina* following a second-step open abdominal uterovaginal anastomosis similar to the push-through technique. (A/B): Method A/B: initial one-step technique analogous to method A and a secondary open abdominal uterovaginal anastomosis analogous to method B.

**Figure 2 jcm-11-05026-f002:**
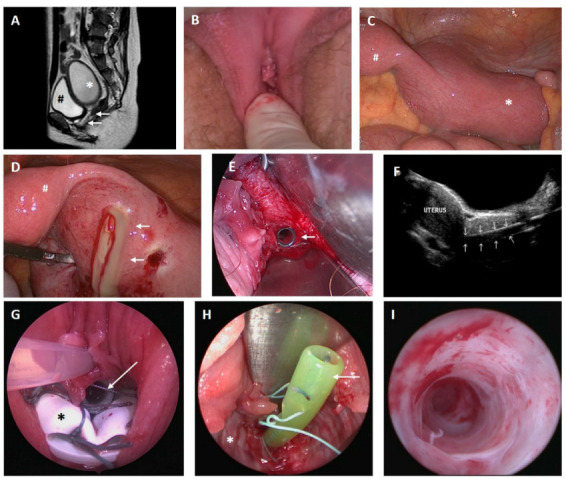
Case 2. (**A**) Magnetic resonance imaging from the pelvis; * hematometra (6.5 × 6.1 × 8.6 cm); arrows show vaginal atresia; # bladder. (**B**) Introitus vagina with vaginal atresia. (**C**) Laparoscopic view of an enlarged right-sided uterus unicornis (*) and a rudimentary left uteral horn (#). (**D**) Discharge of pus after transfundal incision (arrows); # left rudimentary uteral horn. (**E**) Arrow shows intracervical Fehling tube after vaginal and cervical reconstruction. (**F**) Transabdominal sonography; the uterus is displayed with normal endometrium on the left side; the right side of the picture, the intracervical Fehling tube is indicated by small arrows. (**G**) Vaginal device (*) with intracervical Fehling tube (arrow). (**H**) Fehling tube within the uterovaginal anastomosis (arrow), completely epithelialized neovagina (*). (**I**): Latest hysteroscopy after relief of pyometra and cleavage of adhesions.

**Figure 3 jcm-11-05026-f003:**
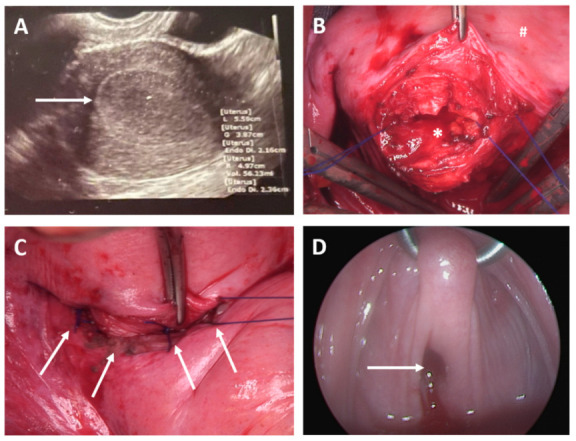
Case 6. (**A**) Transabdominal sonography; uterus with hematometra after weaning from hormonal therapy; arrows show significant hematometra. (**B**) Situs during abdominal laparotomy in preparation for uterovaginal anastomosis similar to the push-through technique; * transverse uterotomy after incision of the plica vesicouterina and caudal preparation of the bladder; # corpus uteri. (**C**) Circular, non-resorbable sutures; Prolene 2.0 establishing uterovaginal anastomosis (arrows). (**D**) Hysteroscopy shows open uterovaginal anastomosis (arrow) 6 weeks after initial surgery.

**Figure 4 jcm-11-05026-f004:**
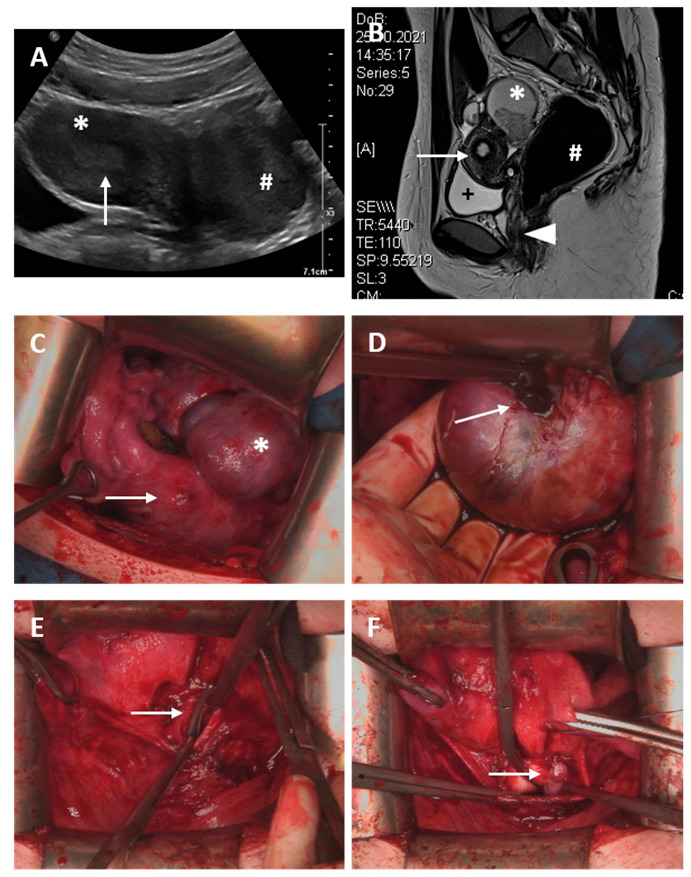
Case 6. (**A**) Transabdominal sonography; * uterus without hematometra; arrow depicts endometrium of 9 mm; # suspected rudimentary horn or myoma. (**B**) Abdominal magnetic resonance imaging: + full bladder, * suspected ovarian cyst 6 × 4 cm, # dilated rectum; arrow shows suspected T-shaped uterus without hematometra; arrowhead shows vagina; no connection from vagina to uterus. (**C**) Situs during abdominal laparotomy: arrow marks T-shaped uterus, * left side ovarian cyst of 10 cm diameter, suspicious for endometrioma (**D**) Incision of the suspected endometrioma and discharge of typical chocolate-like blood (arrow). (**E**): Insertion of an overhold clamp from the vagina (arrow) establishing a connection between the vagina and uterus. (**F**): Insertion of a 14 Ch-silicon catheter (arrow). (**G**) Uterovaginal anastomosis established by circular, non-resorbable Prolene 2.0 sutures (arrow). (**H**) Hysteroscopy after 5 weeks; sufficient uterovaginal anastomosis; * cavum uteri; arrow shows blue non-resorbable Prolene 2.0 suture.

**Table 1 jcm-11-05026-t001:** Patient characteristics.

Case Number	Age at Initial Presentation	ESHRE/ESGE	ASRM	Associated Malformations	Clinical Presentation	Endometriosis at Diagnosis
1	17	U3b C3 V3 / OHVIRA	Uterus didelphis R unilateral cervical agenesis Mid vaginal septum/OHVIRA	Unilateral renal aplasia right side	Hematometra	no
2	13	U4a/C4/V4	R unicornuate uterus with L associated atrophic uterine remnant Cervical agenesis Distal vaginal agenesis	Multiple malformations: double kidney on both sides, skeletal malformations, anal atresia, tetralogy of fallot	Acute abdominal pain, pyometra	no
3	12	U4b C0 V3/4	L unicornuate uterus Normal cervix Distal vaginal septum	Renal agenesis right side	Abdominal pain	no
4	14	U4a C4 V3	R unicornuate with L associated atrophic uterine remnant Cervical agenesis Distal vaginal agenesis	Arterial septum defect and venticel septum defect, Arteria lusoria, konnatal hypothyroidism	Recurrent hematometra after surgery in other clinic	no
5	16	U0 C0 V4	Normal uterus Normal cervix Distal vaginal agenesis	-	Assumed vaginal septum, primary amenorrhea	yes
6	20	U0 C4, V0	Normal Uterus Cervical agenesis Normal vagina	-	Request for treatment, diagnosis at other clinic	yes
7	17	U1a C4 V3/4	T-shaped uterus Cervical agenesis Distal vaginal agenesis	-	Primary amenorrhea	yes
8	11	U0 C4 V0	Normal Uterus Cervical agenesis Normal vagina	-	Primary amenorrhea, pelvic pain	yes

ESHRE/ESGE: European Society of Human Reproduction (ESHRE) and Embryology-European Society for Gynaecological Endoscopy (ESGE); OHVIRA: Obstructed hemivagina with ipsilateral renal anomaly; ASRM: American Society for Reproductive Medicine.

**Table 2 jcm-11-05026-t002:** Surgical outcome parameters.

Case Number	ESHRE/ESGE	Treatment Algorithm	Diagnostic Surgery	Prior Surgery for Neovagina *	Date of Uterovaginal Anastomosis	Number of Surgeries for Uterovaginal Anastomosis	Planned Follow-Up Hysteroscopy And Cervical Dilation	Unplanned Hysteroscopy and Cervical Dilation	Secondary Surgery	Unplanned Hysteroscopy and Cervical Dilation	Total Number of Surgeries
1	U3b C3 V3/OHVIRA	A	0	no	28 April 2015	1	2	0	3	1	7
2	U4a/C4/V4	A	1	no	1 February 2012	1	0	0	1	3	6
3	U4b C0 V3/4	A/B	2 **	no	15 January 2015 15 September 2020	2	1	0	1	5	11
4	U4a C4 V3	A/B	0	no	July 2017 9 July 2019	2	1	1	2	1	7
5	U0 C0 V4	B	0	yes	2 March 2021	1	1	3	0	0	5
6	U0, C4, V0	B	1 **	no	2 February 2022	1	1	0	0	0	3
7	U1a C4 V3/4	B	2 ***	yes **	16 February 2022	1	1	0	0	0	4
8	U0 C4 V0	B	2	no	27 September 2017	1	0	0	0	0	3

OHVIRA: Obstructed hemivagina with ipsilateral renal; * Surgery for neovagina by modified McIndoe technique, vaginal reconstruction or resection of vaginal septum; ** surgery in other hospital, *** one diagnostic surgery in another hospital. (A) Method A: one-step creation of a neovagina * and a simultaneous laparoscopic uterovaginal anastomosis procedure similar to the pull-through technique. (B) Method B: two-step/open abdominal technique and, if necessary, creation a neovagina * following a second-step abdominal uterovaginal anastomosis similar to the push-through technique. (A/B): Method A/B: initial one-step technique analogous to method A, with secondary abdominal uterovaginal anastomosis. Planned follow-up surgery: cervical dilation and sometimes removal of a catheter or tube and hysteroscopy. Minor follow-up surgery: cervical dilation or hysteroscopy. Secondary surgery: procedures other than hysteroscopy and cervical dilation.

**Table 3 jcm-11-05026-t003:** Clinical outcome parameters.

Case Number	Date of Uterovaginal Anastomosis	Latest Examination	Vaginal Lenght	Sexual Intercourse	Orthograde Menstruation	Absence of Hematometra	Re-Stenosis and/ or Hematometra	Hysterectomy	Complications
1	28 April 2015	7 January 2019	n *	lost to follow up	normal menstruation normal left sided hemi uterus and cervix	yes, but pyometra right uterus	recurrent pyometra right hemiuterus	right sided hemi-hysterectomy, preserving the left-sided functional hemiuterus	-
2	26 January 2012	29 October 2018	n *	satisfying	yes, regular cycle	yes	yes, 3 × surgery for vaginal and/or cervical dilation, pyokolpos, sactosalpinx	-	-
3	15 January 2015 15 September 2020	11 April 2021	n *	satisfying	no, hormonal therapy	yes	recurrent vaginal discharge	-	-
4	July 2017 9 July 2019	14 March 2022	n *	not to date	yes	yes	-	-	-
5	2 March 2021	3 January 2022	n *	not to date	pending	yes	yes, 3 × dilation	-	-
6	2 February 2022	15 March 2022	n **	satisfying	pending	yes	pelvic pain and hematormetra after weaning from hormonal contraception	-	-
7	16 February 2022	22 March 2022	n **	not to date	pending	yes	-	-	
8	27 September 2017	30 October 2017	n **	lost to follow up	lost to follow-up	yes	-	-	-

n = normal; * >6 cm counted as normal vaginal length [17]; ** no surgical intervention.

## Data Availability

Patient information concerning medical history, diagnosis, symptoms and treatment were obtained from medical records within the patient data system SAP GUI 7.70_DE as of January 2021 from the Department for Women’s Health of the University of Tuebingen, Germany.

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
