# Peer review of "Reassessment of Surgical Procedures for Complex Obstructive Genital Malformations: A Case Series on Different Surgical Approaches"

_jcm, 2022, doi:10.3390/jcm11175026_

Round 1

Reviewer 1 Report

1. The date of the "date of uterovaginal anastomosis" in case 4 of Table 3 is wrong.

2. Because the follow-up time for cases 6 and 7 is too short, Table 2 cannot faithfully present the patient's actual surgical outcome. Authors should set an appropriate time for post-treatment evaluation so that readers can understand the prognosis of different treatments.

3. The author concluded that the two-step technique was superior. However, this result cannot be objectively presented in this manuscript. (Both cases 5 and 7 are anomalies with V4, but since the follow-up time is too short to know if further surgery will be required like cases 3 and 4)

Reviewer 2 Report

Methods: 

1) How did you choose your patients? All or some?  Better describe decision around treatment approach BEFORE the discussion section.  

     A. More clearly define the inclusion criteria for patients

     B. More clearly define the "McIndoe" technique or neovaginal approach (i.e, skin grafts v buccal mucosa, etc)

2) Please consider changing nomenclature to A and B technique with C eliminated and characterizing those as "failures or complications" of A.

3) Table 1 and 2 could be combined with the addition of the ASRM classification (if desire US readership), prior treatments, other predictors of complications (i.e infection, etc) with better characterization of planned and unplanned procedures.   

4) More clearly recognize and discuss the 50% failure rate of treatment A, with better description if that was how the change in approach occurred over the years.  

5) Simplify and more succinctly describe the cases, with a standardize description/table for each/all (ie, cervix involved y/n)

6) Conclusion of the two step approach being "superior" seems to be a reach, until additional follow up data is available.  Perhaps citing the single step approach as undesirable due to 50% complication or failure rate as unacceptable, leading to changes in approach to care in your institution.  

7) Strongly believe that with 3/8 presented cases having less than 6 months of follow up is premature to adequately outcomes or valid comparisons.  For this reason, I suggest continued follow up and resubmission after suggested changes and longer outcome data.  
